# A Clinical Analysis of Anti-Programmed Death-Ligand 1 (PD-L1) Immune Checkpoint Inhibitor Treatments Combined with Chemotherapy in Untreated Extensive-Stage Small-Cell Lung Cancer

**DOI:** 10.3390/vaccines12050474

**Published:** 2024-04-29

**Authors:** Ping-Chih Hsu, Bing-Chen Wu, Chin-Chou Wang, Li-Chung Chiu, Chiung-Hsin Chang, Ping-Chi Liu, Chiao-En Wu, Scott Chih-Hsi Kuo, Jia-Shiuan Ju, Allen Chung-Cheng Huang, Yu-Ching Lin, Cheng-Ta Yang, How-Wen Ko

**Affiliations:** 1Division of Thoracic Medicine, Department of Internal Medicine, Chang Gung Memorial Hospital at Linkou, Taoyuan City 33305, Taiwan; 8902049@gmail.com (P.-C.H.); tim86112@gmail.com (B.-C.W.); pomd54@cgmh.org.tw (L.-C.C.); joangrace003@gmail.com (C.-H.C.); chihhsikuo@gmail.com (S.C.-H.K.); b9502008@cgmh.org.tw (J.-S.J.); mr0818@cgmh.org.tw (A.C.-C.H.);; 2Department of Medicine, College of Medicine, Chang Gung University, Taoyuan City 33302, Taiwanjiaoen@gmail.com (C.-E.W.); lin0927@cgmh.org.tw (Y.-C.L.); 3Division of Pulmonary & Critical Care Medicine, Kaohsiung Chang Gung Memorial Hospital, Kaohsiung City 83301, Taiwan; 4Department of Internal Medicine, Taoyuan Chang Gung Memorial Hospital, Taoyuan City 33378, Taiwan; 5Division of Pulmonary, Critical Care and Sleep Medicine, Chang Gung Memorial Hospital, Keelung 20401, Taiwan; ewind14@hotmail.com; 6Division of Hematology-Oncology, Department of Internal Medicine, Chang Gung Memorial Hospital at Linkou, Taoyuan City 33305, Taiwan; 7Division of Thoracic Oncology, Department of Respiratory and Critical Care Medicine, Chang Gung Memorial Hospital, Chiayi Branch, Chiayi 613016, Taiwan; 8Department of Respiratory Care, Chang Gung University of Science and Technology, Chiayi Campus, Chiayi 613016, Taiwan; 9Department of Respiratory Therapy, College of Medicine, Chang Gung University, Taoyuan City 33302, Taiwan

**Keywords:** small-cell lung cancer (SCLC), programmed death-ligand 1 (PD-L1), atezolizumab, durvalumab, immunotherapy

## Abstract

Real-world clinical experience of using anti-programmed death-ligand 1 (PD-L1) immune checkpoint inhibitors (ICIs) combined with chemotherapy in the first-line treatment of extensive-stage small-cell lung cancer (SCLC) patients has rarely been reported. In this study, we aimed to perform a retrospective multicenter clinical analysis of extensive-stage SCLC patients receiving first-line therapy with anti-PD-L1 ICIs combined with chemotherapy. Between November 2018 and March 2022, 72 extensive-stage SCLC patients receiving first-line atezolizumab or durvalumab in combination with chemotherapy, according to the cancer center databases of Linkou, Chiayi, and Kaohsiung Chang Gung Memorial Hospitals, were retrospectively included in the analysis. Twenty-one patients (29.2%) received atezolizumab and fifty-one (70.8%) received durvalumab. Objective response (OR) and disease control (DC) rates of 59.7% and 73.6%, respectively, were observed with first-line ICI plus chemotherapy. The median progression-free survival (PFS) was 6.63 months (95% confidence interval (CI), 5.25–8.02), and the median overall survival (OS) was 16.07 months (95% CI, 15.12–17.0) in all study patients. A high neutrophil-to-lymphocyte ratio (NLR; >4) and a high serum lactate dehydrogenase (LDH) concentration (>260 UL) were identified as independent unfavorable factors associated with shorter OS in the multivariate analysis. Regarding safety, neutropenia was the most common grade 3 treatment-related adverse event (AE), but no treatment-related deaths occurred in the study patients. First-line anti-PD-L1 ICIs combined with chemotherapy are effective and safe for male extensive-stage SCLC patients. Further therapeutic strategies may need to be developed for patients with unfavorable outcomes (e.g., baseline high NLR and serum LDH level).

## 1. Introduction

Lung cancer is a major cause of cancer-related death in both males and females worldwide [1,2]. Small-cell lung cancer (SCLC) is a subtype of lung cancer that accounts for approximately 15% of primary lung cancers [1,2]. Among all histological types of lung cancer, SCLC exhibits the most rapid growth and spread to distant sites. Therefore, the majority of SCLC patients exhibit extensive-stage (metastatic) disease at initial diagnosis and have the worst prognosis among all histological types of lung cancer [3,4,5]. Platinum-based chemotherapy in combination with etoposide regimens has been used as a standard first-line therapy for extensive-stage SCLC patients, and these combination regimens provide an objective response rate (ORR) of approximately 50% and a median progression-free survival (PFS) of 4–6 months. The overall survival (OS) of extensive-stage SCLC patients receiving first-line platinum-based therapy combined with etoposide is approximately 8–10 months [4,5,6]. The development of new drugs for the treatment of extensive-stage SCLC has not successfully advanced in the past 3 decades, despite the use of regimens of platinum-based agents combined with etoposide.

Immunotherapies blocking the checkpoint programmed cell death protein 1 (PD-1)/programmed death-ligand 1 (PD-L1) axis have been developed and widely used as anti-cancer therapies for various cancers in the past decade [7,8]. Several pivotal clinical trials have shown that anti-PD1/PD-L1 immune checkpoint inhibitors (ICIs), including pembrolizumab, nivolumab, atezolizumab, and durvalumab, significantly improve the survival of advanced non-small-cell lung cancer (NSCLC) patients when compared with conventional chemotherapy [9,10]. The efficacy of anti-PD1/PD-L1 ICIs combined with chemotherapy as a front-line therapy for extensive-stage SCLC has been explored in prospective clinical trials [11,12,13]. Atezolizumab and durvalumab are the first two ICIs to be approved for combination with chemotherapy in first-line therapy for extensive-stage SCLC, based on the promising results of two pivotal clinical trials (IMpower133 and CASPIAN) [14,15], both of which showed that atezolizumab or durvalumab in addition to chemotherapy significantly improved the OS of extensive-stage SCLC patients compared with chemotherapy alone.

Although atezolizumab or durvalumab in combination with chemotherapy has been suggested as a first-line regimen for the treatment of extensive-stage SCLC, the anti-PD-L1 ICIs are not widely covered by public health insurance policies in most countries, due to concerns about cost-effectiveness [16,17]. The use of atezolizumab or durvalumab is only affordable in a few extensive-stage SCLC patients outside of clinical trials [16,17]. Therefore, the real-world clinical outcomes of extensive-stage SCLC patients receiving combination therapies have rarely been reported, and more clinical experiences of using combination therapies need to be explored. In this study, we aimed to conduct a retrospective multicenter observational study to investigate the efficacy and safety of atezolizumab and durvalumab combined with chemotherapy as first-line treatments for extensive-stage SCLC patients in real-world clinical practice.

## 2. Materials and Methods

### 2.1. Patients, Treatment, and Follow-Up

The study patients were retrospectively reviewed from the cancer center databases of Linkou, Keelung, Chiayi, and Kaohsiung Chang Gung Memorial Hospitals (CGMHs). A total of 79 histologically diagnosed and untreated extensive-stage SCLC patients who had received ICIs as first-line therapy between November 2018 and March 2023 were screened, and 72 patients were ultimately included in the analysis. The inclusion criteria for the study subjects were as follows: (1) received the ICI atezolizumab or durvalumab, (2) received atezolizumab or durvalumab in combination with chemotherapy, and (3) received at least 2 cycles of ICI combined with chemotherapy. Patients were excluded from the analysis for the following reasons: (1) received the ICI pembrolizumab, (2) received ICI therapy alone without combination chemotherapy, or (3) received only one cycle of combination therapy. The electronic charts of all 79 patients were screened and reviewed by at least 2 of our authors, and 72 patients ultimately met the inclusion criteria of this study. The clinical variables and treatment information from the electronic charts of these 72 study patients were reviewed and recorded for analysis by at least 2 of our authors. The patients included in this study are summarized in Figure 1.

All of the study patients received contrast-medium-enhanced computed tomography (CT), magnetic resonance imaging (MRI) of the brain, and whole-body fluorodeoxyglucose (FDG) positron emission tomography (PET) scans to verify the baseline stages. All patients in this study underwent whole-body CT scans roughly every 3 to 4 months during treatment, in order to evaluate their disease status and treatment response. Additional imaging tests, such as chest plain film, sonograms, MRI, bone scans, and FDG-PET scans, were ordered as needed by clinical physicians during treatment and follow-up to assist in judging the disease status.

The baseline clinical factors of neutrophil-to-lymphocyte ratio (NLR), serum lactate dehydrogenase (LDH) level, and serum Na concentration level were measured before the first cycle of anti-PD-L1 ICIs combined with chemotherapy.

Treatment response in this study was evaluated by using the Response Evaluation Criteria in Solid Tumors (RECIST) version 1.1, and responses were classified as complete response (CR), partial response (PR), stable disease (SD), or progressive disease (PD). The progression-free survival (PFS) duration was defined from the first date of treatment administered to the date of PD, detected by imaging or determined by clinical physicians. The duration of OS was defined from the first date of treatment administered to the date of death. If patients were still alive through the last follow-up date (December 31, 2023), PFS and OS were censored at the date of the last clinical visit. Treatment-related adverse events (AEs) were recorded and graded using the National Cancer Institute’s Common Terminology Criteria. The symptoms and management of these AEs recorded in electronic patient charts were carefully reviewed for analysis.

### 2.2. Statistical Analysis

The demographic characteristics and treatment information in this study are presented as quantitative variables. The age, NLR, serum LDH levels, and serum Na concentration levels of the study patients are presented as medians and ranges. We used univariate and multivariate Cox regression analyses of OS according to different clinical factors. K–M survival curves were generated to compare OS between the groups with different factors. The OS survival curves were generated using GraphPad Prism (version 5.0; GraphPad Software, San Diego, CA, USA). Two-sided *p*-values less than 0.05 were considered to indicate statistical significance. IBM SPSS Statistics version 22.0 (SPSS Corp., Chicago, IL, USA) was used for the statistical analysis.

## 3. Results

### 3.1. Baseline Demographic Characteristics and Clinical Information of the Study Patients

The baseline demographic characteristics and clinical information of the 72 patients in this study are shown in Table 1. Most of the patients in this study were male (95.8%) and had a history of smoking (98.6%). The median age of the study patients was 65 years (range: 39 to 86). The median body mass index (BMI) of the study patients was 23.81 (range: 18.62 to 31.33). Regarding the sites of distant metastasis at initial diagnosis, 26 (36.1%) had contralateral lung-to-lung metastases, 28 (38.9%) had pleural metastases, 5 (6.9%) had adrenal metastases, 11 (15.3%) had brain metastases, 22 (30.6%) had bone metastases, and 15 (20.8%) had liver metastases. Twenty-one (29.2%) patients received atezolizumab as their first-line therapy, while the other fifty-one (70.8%) received durvalumab. Hyponatremia (serum NA < 130 meq/L) was recorded in 12 (16.7%) of the study patients, suggesting that these patients may have had a paraneoplastic syndrome of inappropriate antidiuretic hormone (SIADH) or increased ectopic ACTH secretion. Regarding the chemotherapy regimens administered in this study, 49 (68.1%) patients received cisplatin plus etoposide, and 23 (31.9%) patients received carboplatin plus etoposide. All 11 (15.3%) brain metastatic patients in this study received radiation therapy for the treatment of brain metastases, and 6 (8.3%) patients received radiation therapy for bone metastases. The radiation therapies administered during the first-line combination therapy were for the purpose of palliative treatment. Four patients (5.6%) received prophylactic cranial irradiation (PCI) after chemotherapy, and all of the patients received six cycles of chemotherapy and achieved a PR to first-line treatment.

### 3.2. Efficacy of First-Line Anti-PD-L1 ICIs Combined with Chemotherapy

Among the 72 patients who received first-line therapy consisting of anti-PD-L1 ICIs combined with chemotherapy, 43 (59.7%) had a PR, 10 (13.9%) had SD, and 19 (26.4%) had PD. The ORR and disease control rate (DCR) were 59.7% and 73.6%, respectively (Figure 2a). The median PFS was 6.63 months (95% confidence interval (CI), 5.25–8.02; Figure 2b), and the median OS was 16.07 months (95% CI, 15.12–17.0; Figure 2c) for all patients.

### 3.3. Analysis of Predictive Factors Associated with OS

The median OS according to different clinical variables was analyzed via Cox regression, and the results are shown in Table 2. The age of 65 was defined as a cutoff value, as the median age of all patients in the study was 65. According to the univariate analysis, female sex, a high NLR, high serum LDH levels, and AEs of grade 3 neutropenia were significantly associated with shorter OS. The factors of distant metastatic sites in the brain, bone, and liver were not shown to be significantly associated with OS in this study. According to the multivariate analysis, a high NLR and high serum LDH levels were identified as independent predictors of unfavorable OS. Patients were categorized into different NLR and LDH levels to compare their OS using Kaplan–Meier survival curves. In this study, patients with a lower NLR (≤4) had significantly longer OS than those with a high NLR (>4; 17.33 vs. 5.38 months, hazard ratio (HR) = 0.143; CI = 0.068–0.299, *p* < 0.001; Figure 3a). Patients with lower serum LDH levels (≤260 UL) had significantly longer OS than those with high serum LDH levels (>260 UL; 20.4 vs. 5.8 months, HR = 0.143; CI = 0.094–0.358, *p* < 0.001; Figure 3b).

### 3.4. First-Line-Treatment-Related Adverse Events (AEs)

The treatment-related AEs experienced by patients receiving first-line immunotherapy combined with chemotherapy are shown in Table 3, and they are classified as non-hematological or hematological toxicity. Among non-hematological toxicities, the most frequent AE was nausea and vomiting (66.7%), followed by hair loss (54.2%), fatigue (36.1%), diarrhea (18.1%), abnormal renal function (18.1%), increased ALT (16.7%), skin rashes (16.7%), increased AST (15.3%), stomatitis (13.9%), and constipation (2.8%). Only one patient experienced grade 3 diarrhea, which also led to grade 3 elevated serum creatinine. This patient had been hospitalized to receive intravenous fluid supplementation, which corrected the complication of acute kidney injury. This patient did not experience severe diarrhea and kidney injury in the course of subsequent treatment. Most grade 3 AEs observed in this study were hematological toxicities. The most common grade 3 hematological AE in this study was neutropenia (22.2%), which led to the treatments being postponed. Twenty-two (30.5%) patients in this study experienced febrile neutropenia, and all of the patients received granulocyte colony-stimulating factor (G-CSF) and antibiotics due to concerns about infection. Skin rashes were considered to be an immune-mediated adverse event (imAE); no other severe imAEs—such as pneumonitis, cardiomyopathy, or endocrinopathies—were recorded in this study. Overall, the safety of first-line immunotherapy combined with chemotherapy was manageable, and no treatment-related deaths were recorded in this study.

A comparison of anti-PD-L1 ICIs with chemotherapy between this study and the prospective trials IMpower133 and CASPIAN is shown in Table 4.

## 4. Discussion

To the best of our knowledge, this study is the first to report the use of first-line anti-PD-L1 ICIs (atezolizumab and durvalumab) combined with standard chemotherapy for the treatment of extensive SCLC in real-world clinical practice. The use of atezolizumab or durvalumab in combination with platinum plus etoposide regimens as first-line therapy provided a 59.7% ORR and 6.63 months of PFS. Patients receiving first-line atezolizumab or durvalumab combined with chemotherapy in this study had a median OS of 16.07 months. We found that a high NLR and serum LDH concentration were independent predictive factors associated with shorter OS in the studied patients. Hematological AE neutropenia accounted for most of the treatment-related AEs in this study; these AEs were manageable, and no treatment-related deaths occurred.

In the clinical trial IMpower133, the chemotherapy regimen used was carboplatin plus etoposide in all patients [14]. In the other clinical trial, CASPIAN, 78% of the study’s patients received carboplatin plus etoposide as a chemotherapy regimen [15]. In two previous retrospective studies, the treatment regimens were atezolizumab combined with carboplatin plus etoposide [18,19]. In contrast to these previous prospective clinical trials and retrospective studies, most of the patients (68.1%) in our study received regimens of cisplatin plus etoposide. In Taiwan, the Health Insurance Agency mainly reimburses the chemotherapy regimen cisplatin, rather than carboplatin, for the treatment of small-cell lung cancer [20], which explains why most of the patients in our study received cisplatin as chemotherapy. Fifty-one patients (70.8%) in this study received durvalumab as immunotherapy, but real-world experience using durvalumab combined with chemotherapy as first-line therapy in extensive SCLC has rarely been reported [21]. Our results showed that patients who received first-line treatment with atezolizumab or durvalumab in combination with chemotherapy experienced no significant difference in OS. The chemotherapy regimen of cisplatin combined with etoposide is the standard first-line treatment for extensive-stage SCLC [22]. The results of our study indicate that the use of atezolizumab or durvalumab combined with standard chemotherapy regimens as first-line therapy in extensive-stage SCLC is effective and feasible in clinical practice.

The NLR has been identified as a predictive factor associated with poor clinical outcomes in patients with NSCLC and SCLC in several previous studies [23,24,25]. An increasing NLR has been proposed to be a response to systemic inflammation, and in previous studies it has been shown to be correlated with the severity of trauma, CVD, and malignancies. These previous studies demonstrated that increasing the NLR negatively affects survival in patients with trauma and cardiovascular events [26,27]. In our study, the NLR was suggested to be a factor associated with the prognosis of extensive-stage SCLC patients receiving anti-PD-L1 ICIs combined with chemotherapy, but it may not be a biomarker of the treatment response to such therapy. Two previous studies reported that elevated NLR was not identified as an independent factor associated with OS in limited-stage SCLC and resectable NSCLC [28,29]; in contrast to these two studies, all of our study’s patients were at metastatic stages, rather than limited or early stages. A previous study showed that an elevated NLR was associated with increased risk of disease relapse and metastasis in head-and-neck squamous-cell carcinoma [27]. Patients who had ever experienced treatment-related grade 3 neutropenia were found to have significantly shorter OS than those without grade 3 neutropenia in this study. Another previous study reported that induction chemotherapy reduced the neutrophils and inflammatory cytokines in the cancer microenvironment and increased the anti-cancer efficacy of anti-PD-L1 ICIs. The administration of induction chemotherapy was found to decrease the NLR and improve the outcomes of cancer patients receiving anti-PD-L1 immunotherapy [30]. However, severe myelosuppression during anti-cancer therapy has been reported to negatively affect the survival outcomes of cancer patients, because severe myelosuppression (such as neutropenia) leads to the delay, interruption, and dose reduction of anti-cancer treatments [31]. Taken together, these factors explain why patients with grade 3 neutropenia had significantly shorter OS than those without severe neutropenia. 

The serum LDH concentration is another prognostic factor that is frequently identified in patients with malignancies [19,23,24,32,33]. LDH is an essential enzyme in glycolysis and functions through catalyzing the conversion of pyruvate to lactate. Elevated serum LDH levels have been used as an inflammatory index [34]. The Warburg effect refers to the ability of cancer cells to derive energy through glycolysis even under aerobic conditions. Therefore, elevated serum LDH levels may reflect increased cancer cell activation and are associated with poor prognosis in cancer patients [34]. The results of our study are compatible with those of previous studies [19,23,24,32,33,34,35], and LDH was identified as a factor associated with unfavorable outcomes in extensive-stage SCLC patients who received first-line anti-PD-L1 ICIs combined with chemotherapy. The three female patients in this study had significantly shorter OS than the male patients. All three female patients had high NLRs (9–11) and serum LDH levels (770–1767 UL), which were not shown in the Results section. Therefore, the factors associated with shorter OS in these three patients were elevated NLRs and serum LDH levels, unrelated to the factor of sex. 

Previous studies have reported that additional radiation therapy induces abscopal effects and enhances the cytotoxicity of ICIs in the treatment of NSCLC [36,37]. Based on the concept of an abscopal effect, additional thoracic radiation therapy was administered to some extensive-stage SCLC patients receiving immunotherapy combined with chemotherapy in previous studies [18,23,24]. Superior vena cava (SVC) syndrome and a large thoracic tumor burden are paraneoplastic complications that frequently occur in extensive-stage SCLC patients, and thoracic radiation therapy must be administered to address the complications of great vessels or airway compression [38,39]. No patient in our study received thoracic radiation therapy during first-line treatment, and the efficacy of first-line immunotherapy combined with chemotherapy in our study was not inferior to that determined in previous studies. The OS outcomes of our study patients were also consistent with those of previous studies [18,19,23,24,25]. The analysis of several previous studies investigating the use of anti-PD-L1 ICIs in extensive-stage SCLC showed that administering thoracic radiation was not positively associated with OS [23,24,25]. Taking these results together, additional thoracic radiation therapy may have a limited beneficial effect on treatment efficacy or clinical outcomes and should be reserved for patients with paraneoplastic complications such as SVC syndrome.

Hyponatremia-induced SIADH and ectopic ACTH secretion are paraneoplastic syndromes that commonly occur in SCLC patients, the incidence rates of which are 10–15%. Twelve (16.7%) patients in this study were recorded as having hyponatremia at diagnosis, and this result indicates that the incidence of SIADH or ectopic ACTH secretion was consistent with previous studies [40,41]. The AEs of SIADH and ectopic ACTH secretion are induced by the disease SCLC itself, unrelated to the treatments. Immune-mediated pneumonitis is an imAE and a major morbidity in cancer patients who receive anti-PD-1/PD-L1 ICIs. Although the incidence rate of immune-mediated pneumonitis is less than 5% in lung cancer patients receiving anti-PD-1/PD-L1 immunotherapy, this adverse event is a major morbidity that can lead to permanent discontinuation of immunotherapy and even mortality in severe cases [40,41]. Fortunately, none of our study’s patients experienced immune-mediated pneumonitis. According to an analysis of a previous study, patients who had ever received thoracic radiation therapy had an increased risk of immune-mediated pneumonitis during anti-PD-1/PD-L1 immunotherapy [42,43]. This may explain why none of our study’s patients experienced immune-mediated pneumonitis, because no patient in this study received concurrent thoracic radiation therapy during first-line treatment.

There are several limitations to this study that should be acknowledged. Due to the retrospective nature of the study, the treatments, including chemotherapy, immunotherapy, and radiation therapy, were administered based on the judgments of clinical physicians rather than on the protocol of the clinical trial. All 11 patients with baseline brain metastases in our study received palliative radiation therapy during their first-line treatment. In addition, six patients in this study received palliative radiation therapy for bone metastases. The additional local radiation therapy may have positively affected the survival of patients in this study. In the clinical trials IMpower133 and CASPIAN, patients with symptomatic brain metastases were required to be treated before entering the trials [14,15]. Unlike the prospective clinical trials, some patients in usual clinical practice have symptoms related to brain or bone metastases that need palliative local therapies for symptomatic relief. In the CASPIAN cohort, PCI was administered only in the arm receiving chemotherapy alone, and two patients in this study received PCI after first-line treatment with durvalumab combined with chemotherapy [15]. In addition, AEs were reviewed and analyzed from electronic medical records, and the records of AEs in this study could not be as detailed as those in clinical trials. For example, although stomatitis was noted in this study, the presentations could not be determined as oral pain, sore throat, or mucosal ulceration in the oral cavity. Endocrinopathies such as hypothyroidism and adrenal insufficiency sometimes occur in cancer patients who receive anti-PD-L1 ICIs [44], but we did not find these AEs after retrospectively reviewing the electronic medical records.

Although our study’s patients were drawn from four different institutions, all of these institutions are located in the cities of west Taiwan. The ethnic and geographic characteristics of this study’s patients were homogeneous. In addition, most of the patients in this study were male (95.8%), with only three female patients (4.2%), so our study’s results may be not representative of female extensive-stage SCLC patients. Although the regimens of the combination treatments were not all the same as the regimens used in the two clinical trials (IMpower133 and CASPIAN) [14,15], neither the efficacy nor safety was negatively affected. Based on the results of our study, further clinical studies should focus on the efficacy and safety of anti-PD-L1 ICIs combined with chemotherapy in extensive-stage SCLC patients of female sex, ethnic groups other than East Asian, and those from different geographic areas. 

## 5. Conclusions

First-line anti-PD-L1 ICIs combined with standard chemotherapy are feasible for male extensive-stage SCLC patients, considering their efficacy and safety. Increased baseline NLR and serum LDH levels were associated with unfavorable outcomes, and further therapeutic strategies may be needed for these patients.

## Figures and Tables

**Figure 1 vaccines-12-00474-f001:**
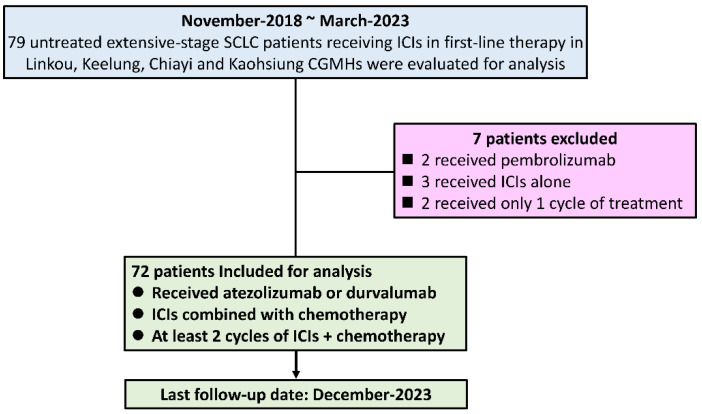
Flow chart of study patient retrieval.

**Figure 2 vaccines-12-00474-f002:**
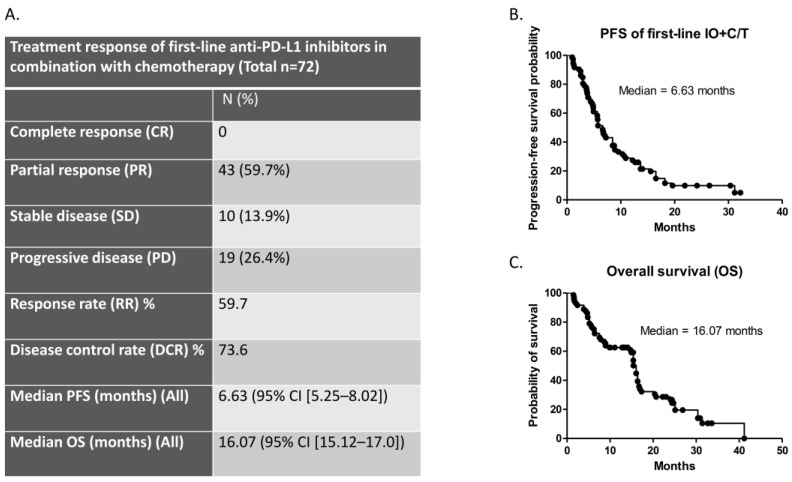
Efficacy of first-line anti-PD-L1 ICIs combined with chemotherapy: (**A**) treatment response to first-line anti-PD-L1 inhibitors in combination with chemotherapy; (**B**) median PFS in patients treated with first-line anti-PD-L1 inhibitors combined with chemotherapy; (**C**) the median OS of all patients.

**Figure 3 vaccines-12-00474-f003:**
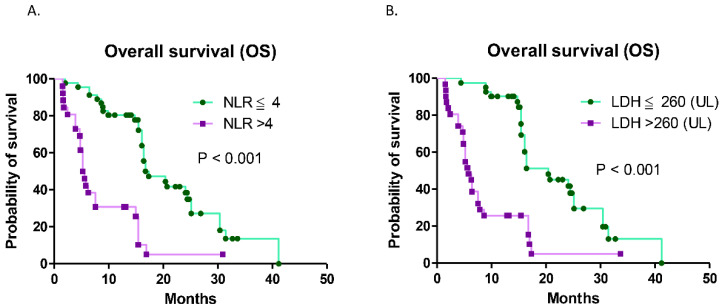
Overall survival (OS) was analyzed based on the neutrophil-to-lymphocyte ratio (NLR) (>4) and serum lactate dehydrogenase (LDH) level (>260 UL) via Kaplan–Meier survival curves: (**A**) comparison of the median OS between patients with different NLRs (HR = 0.143; 95% CI, 0.068–0.299; *p* < 0.001); (**B**) comparison of the median OS between patients with different LDH levels (HR = 0.143; 95% CI = 0.094–0.358; *p* < 0.001).

**Table 1 vaccines-12-00474-t001:** Baseline demographic characteristics of the study patients.

Total	N = 72 (%)
Male	69 (95.8%)
Female	3 (4.2%)
Age year (median/range)	65 (39 – 86)
ECOG PS	
0–1	67 (93.1%)
2	5 (6.9%)
BMI (median/range)	23.81 (18.62–31.33)
Smoking status	
Non-smoker	1 (1.4%)
Former/current smoker	71(98.6%)
Histology	
Small-cell carcinoma	72 (100%)
Stage	
IV	72 (100%)
Neutrophil-to-lymphocyte ratio (NLR) (median/range)	3.3 (1.3–49.0)
LDH (UL) (median/range)	227.0 (150.0–2826.0)
Na (meq/L) (median/range)	136.0 (115–149)
Na < 130 meq/L	12 (16.7%)
Distant metastatic sites	
Contralateral lung	26 (36.1%)
Pleura	28 (38.9.1%)
Adrenal gland	5 (6.9%)
Brain metastasis	11 (15.3%)
Bone metastasis	22 (30.6%)
Liver metastasis	15 (20.8%)
Anti-PD-L1 inhibitors	
Atezolizumab	21 (29.2%)
Durvalumab	51 (70.8%)
Cycles of IO (median/range)	8 (2–29)
Chemotherapy regimens	
Cisplatin + etoposide	49 (68.1%)
Carboplatin + etoposide	23 (31.9%)
Local radiation during IO + chemotherapy	
Brain	11 (15.3%)
Bone	5 (8.3%)
Prophylactic cranial irradiation (PCI) after chemotherapy	4 (5.6%)

Abbreviations: BMI, body mass index; ECOG PS, Eastern Cooperative Oncology Group performance status; PD-L1, programmed death-ligand 1.

**Table 2 vaccines-12-00474-t002:** Cox regression of predictive factors associated with overall survival (OS).

Variables	Patients N (%)	Median OS (Months)	Univariate Analysis *p*-Value HR (95% CI)	Multivariate Analysis	
				HR (95% CI)	*p*-Value
**Age**					
≤65 years	38	15.40	0.763		
>65 years	34	16.07	0.813 (0.462–1.432)		
**Sex**					
Male	69	16.07	0.01		
Female	3	4.83	0.10 (0.027–0.370)		
**ECOG PS**					
0–1	67	16.07	0.075		
2	5	8.97	2.337 (0.917–5.959)		
**NLR**					
≦4	46	17.33	<0.001	1	
>4	26	5.20	4.084 (2.291–7.283)	2.994 (1.607–5.578)	0.001
**LDH (UL)**					
≦260	41	20.4	<0.001	1	0.001
>260	31	5.8	3.802 (2.159–7.283)	2.921 (1.591–5.362)	
**Neutropenia**					
Without grade 3	56	16.4	0.008		
With grade 3	16	5.20	2.52 (1.337–4.739)		
**Anti-PD-L1 inhibitors**					
Atezolizumab	21	15.4	0.115		
Durvalumab	51	16.4	0.618 (0.344–1.108)		
**Metastatic sites**					
**Brain**					
With brain metastasis	11	16.1	0.713		
Without brain metastasis	61	16.4	1.140 (0.568–2.287)		
**Bone**					
With bone metastasis	22	15.4	0.099		
Without bone metastasis	50	16.1	1.630 (0.922–2.883)		
**Liver**					
With liver metastasis	15	15.4	0.606		
Without liver metastasis	57	16.7	1.179 (0.631–2.201)		

Abbreviations: ECOG PS, Eastern Cooperative Oncology Group performance status; NLR, neutrophil-to-lymphocyte ratio; HR, hazard ratio; EGFR, epidermal growth factor receptor; CI, confidence interval.

**Table 3 vaccines-12-00474-t003:** Treatment-related adverse events (AEs) associated with anti-PD-L1 inhibitors in combination with chemotherapy.

Adverse Events (AEs)	All *n* = 72	Grade 1–2, n (%)	Grade 3, n (%)	Grade 4, n (%)
Non-hematological				
Nausea or vomiting	48 (66.7%)	48 (66.7%)	0 (0%)	0 (0%)
Diarrhea	13 (18.1%)	12 (16.7%)	1 (1.4%)	0 (0%)
Constipation	2 (2.8%)	2 (2.8%)	0 (0%)	0 (0%)
Stomatitis	10 (13.9%)	10 (13.9%)	0 (0%)	0 (0%)
Fatigue	26 (36.1%)	26 (36.1%)	0 (0%)	0 (0%)
Skin rashes	12 (16.7%)	12 (16.7%)	0 (0%)	0 (0%)
Hair loss	39 (54.2)	39 (54.2%)	0 (0%)	0 (0%)
Increased liver transaminases				
Increased AST	11 (15.3%)	11 (15.3%)	0 (0%)	0 (0%)
Increased ALT	12 (16.7%)	12 (16.7%)	0 (0%)	0 (0%)
Increased creatinine	13 (18.1%)	12 (16.7%)	1 (1.4%)	0 (0%)
Hematological				
Neutropenia	58 (80.5%)	42 (58.3%)	16 (22.2%)	0 (0%)
Anemia	59 (81.9%)	46 (63.9%)	13 (18.0%)	0 (0%)
Thrombocytopenia	47 (65.2%)	41 (56.9%)	6 (8.3%)	0 (0%)
	Any grade			
Febrile neutropenia	22 (30.5%)			

Abbreviations: AE, adverse event.

**Table 4 vaccines-12-00474-t004:** Comparison of anti-PD-L1 ICIs with chemotherapy between this study and the prospective trials IMpower133 and CASPIAN.

Study Name	Hsu et al.	IMpower133 [14]	CASPIAN [15]
Ant-PD-L1 ICIs	Atezolizumab/ durvalumab	Atezolizumab	Durvalumab
Chemotherapy regimens	Cis/carbo + etoposide	Carbo + etoposide	Cis/carbo + etoposide
Chemotherapy cycles	4–6	4 cycles for all arms	4 cycles for IO arms
Brain metastasis	15.3%	8.6%	10.2%
Liver metastasis	20.8%	37%	39.5%
ORR, % IO + chemotherapy	59.7%	60.2%	68.0%
Median PFS, months	6.63	5.2	5.1
Median OS, months	16.07	12.3	13.0
>Grade 3 AE, % (any)	51.3%	57.1%	52%
Treatment-related death	0	1.5%	2.3%

## Data Availability

The data sets generated and analyzed during this study are not publicly available because of local regulations regarding medical confidentiality.

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
