# Peer review of "A Clinical Analysis of Anti-Programmed Death-Ligand 1 (PD-L1) Immune Checkpoint Inhibitor Treatments Combined with Chemotherapy in Untreated Extensive-Stage Small-Cell Lung Cancer"

_vaccines, 2024, doi:10.3390/vaccines12050474_

Round 1

Reviewer 1 Report

Comments and Suggestions for Authors

Hsu et. al. describe a retrospective analysis on records of 72 patients treated for extensive-stage SCLC with cisplatin and anti-PD-L1 inhibitors at two different Taiwan hospitals.  This analysis demonstrates that these treatments do not increase previously observed toxicities described in clinical trials, while demonstrating measurably better results.  Also as shown for other cancers, high neutrophil to lymphocyte ratios and elevated LDH serum level correlated with poorer outcomes.

The paper is very well written, and the data are appropriately presented. The conclusions are consistent with the data and therefore convincing.  A few points can be addressed that may improve the paper somewhat.

What is meant by “real world”?  Is it “not a clinical trial”? This could be stated in the introduction, however, if this is a commonly understood term in the clinical realm, then perhaps a definition would not be necessary.

Since the study is highly skewed towards men, perhaps the conclusion should be updated to say that this is a safe treatment for men instead of patients in general.

The three females who were part of the study certainly are not powered sufficiently to make statistical conclusions, but the fact that their survival was very different vs the 69 males should merit a brief mention in the discussion – if only perhaps to say that this indicates further study in females is warranted.  It would also be nice to know where the females fell in the NLR and LDH data.

Are the 22 patients who suffered neutropenia part of the higher surviving group with the low NLR? If so, is induction of neutropenia a good thing?

A table comparing the findings of this retrospective study against the data that were found in the clinical trials would be very nice to see.

Author Response

Response to Reviewer 1 Comments

Hsu et. al. describe a retrospective analysis on records of 72 patients treated for extensive-stage SCLC with cisplatin and anti-PD-L1 inhibitors at two different Taiwan hospitals.  This analysis demonstrates that these treatments do not increase previously observed toxicities described in clinical trials, while demonstrating measurably better results.  Also as shown for other cancers, high neutrophil to lymphocyte ratios and elevated LDH serum level correlated with poorer outcomes.

The paper is very well written, and the data are appropriately presented. The conclusions are consistent with the data and therefore convincing.  A few points can be addressed that may improve the paper somewhat.

Point 1: What is meant by “real world”?  Is it “not a clinical trial”? This could be stated in the introduction, however, if this is a commonly understood term in the clinical realm, then perhaps a definition would not be necessary.

Response 1: In response to this point, we revised the title of manuscript as “ A Clinical Analysis of Anti-Programmed Death-Ligand 1 (PD-L1) Im-mune Checkpoint Inhibitor Treatments Combined with Chemothera-py in Untreated Extensive Stage Small Cell Lung Cancer” as suggested.

Point 2: Since the study is highly skewed towards men, perhaps the conclusion should be updated to say that this is a safe treatment for men instead of patients in general

Response 2: In response to the concern about “the study is highly skewed towards men”, we had revised our conclusion of this manuscript as suggested.

We added a paragraph in the discussion section to address this comment.

Most of the patients in this study were male (95.8%), with only three female patients (4.2%), so our study’s results may be not representative of female extensive-stage SCLC patients. Based on the results of our study, further clinical studies should focus on the efficacy and safety of anti-PD-L1 ICIs combined with chemotherapy in extensive-stage SCLC patients of female sex, ethnic groups other than East Asian, and from different geo-graphic areas.

Point 3: The three females who were part of the study certainly are not powered sufficiently to make statistical conclusions, but the fact that their survival was very different vs the 69 males should merit a brief mention in the discussion – if only perhaps to say that this indicates further study in females is warranted. It would also be nice to know where the females fell in the NLR and LDH data.

Response 3:

We had added a paragraph to discuss this comment in revised manuscript as suggested.

      The three female patients in this study had significantly shorter OS than the male patients. All three female patients had high NLR (9–11) and serum LDH levels (770–1767 UL), which were not shown in the Results section. Therefore, the factors associated with shorter OS in these three patients were elevated NLR and serum LDH levels, un-related to the factor of sex.

Point 4: Are the 22 patients who suffered neutropenia part of the higher surviving group with the low NLR? If so, is induction of neutropenia a good thing? 

Response 4: In response to the comment of “are the 22% patients who suffered neutropenia part of the higher surviving group with the low NLR?”, we had done further analysis and discussion to address this question.

Patients who had ever experienced treatment-related grade 3 neutropenia were found to have significantly shorter OS than those without grade 3 neutropenia in this study. Another previous study reported that induction chemotherapy reduced the neutrophils and inflammatory cytokines in the cancer microenvironment and increased the an-ti-cancer efficacy of anti-PD-L1 ICIs. The administration of induction chemotherapy was found to decrease the NLR and improve the outcomes of cancer patients receiving anti-PD-L1 immunotherapy [1]. However, severe myelosuppression during an-ti-cancer therapy has been reported to negatively affect the survival outcomes of cancer patients, because severe myelosuppression (such as neutropenia) leads to the delay, interruption, and dose reduction of anti-cancer treatments [2]. Taken together, these factors explain why patients with grade 3 neutropenia had significantly shorter OS than those without severe neutropenia.

Point 5: A table comparing the findings of this retrospective study against the data that were found in the clinical trials would be very nice to see.

Response 5: In response to this comment, we added a table 4 to compare our study with trials of IMpower133 and CASPIAN as suggested.

Reference:

  1. Galon, J.; Bruni, D.;. Approaches to treat immune hot, altered and cold tumours with combination immunotherapies. Nat Rev Drug Discov. 2019 Mar;18(3):197-218.

  1. Crawford, J.; Herndon, D.; Gmitter, K.; Weiss, J. The impact of myelosuppression on quality of life of patients treated with chemotherapy. Future Oncol. 2024 Apr 8.

Reviewer 2 Report

Comments and Suggestions for Authors

Comments:

1. The patients of this study were from 4 different hospitals in Taiwan. Is geological difference significant among hospitals?

2. Three female patients were recruited in this study. Even though the female sample size is very small, will the conclusion from this study be different between 3 female patients and 69 male patients?

3. Table 1: please include BMI and other detailed patients' information. In metastatic sites, the combination of brain, bone, and liver metastasis is 66.7%. What are other sites? Also do SCLCs from patients express ACTH or ADH?

4. Table 2: the patient number of ECOG PS is different from the one on Table 1. Please explain. Also why chose age of 60 as cut-off?

5. Table3: it is better to expand the adverse events, such as weakness/fatigability, hair loss, mouth sores, etc. which liver transaminase was measured in the study? 

6. The corresponding author needs to use the institute's email for communication.

Comments on the Quality of English Language

Minor editing of English language required

Author Response

Response to Reviewer 2 Comments

Point 1: The patients of this study were from 4 different hospitals in Taiwan. Is geological difference significant among hospitals? ʉ۬

Response 1: In response to the concern about “Is geological difference significant among hospitals?”, although our study’s patients were drawn from four different institutions, all of these institutions are located in the cities of west Taiwan. The ethnic and geographic characteristics of this study’s patients were homogeneous.

Based on the results of our study, further clinical studies should focus on the efficacy and safety of anti-PD-L1 ICIs combined with chemotherapy in extensive-stage SCLC patients of female sex, ethnic groups other than East Asian, and from different geo-graphic areas.

Point 2: Three female patients were recruited in this study. Even though the female sample size is very small, will the conclusion from this study be different between 3 female patients and 69 male patients?

Response 2: In response to the concern about “will the conclusion from this study be different between 3 female patients and 69 male patients?”, we had revised our conclusion of this manuscript as suggested.

   We added a paragraph in the discussion section to address this comment.

Most of the patients in this study were male (95.8%), with only three female patients (4.2%), so our study’s results may be not representative of female extensive-stage SCLC patients. Based on the results of our study, further clinical studies should focus on the efficacy and safety of anti-PD-L1 ICIs combined with chemotherapy in extensive-stage SCLC patients of female sex, ethnic groups other than East Asian, and from different geo-graphic areas.

Point 3: Table 1: please include BMI and other detailed patients' information. In metastatic sites, the combination of brain, bone, and liver metastasis is 66.7%. What are other sites? Also do SCLCs from patients express ACTH or ADH?

Response 3: We had added the additional information in revised manuscript as suggested in response to this comment.

The median age of the study patients was 65 years (range: 39 to 86). The median body mass index (BMI) of the study patients was 23.81 (range: 18.62 to 31.33). Regarding the sites of distant metastasis at initial diagnosis, 26 (36.1%) had contralateral lung-to-lung metastases, 28 (38.9%) had pleural metastases, 5 (6.9%) had adrenal metastases, 11 (15.3%) had brain metastases, 22 (30.6%) had bone metastases, and 15 (20.8%) had liver metastases.

Hyponatremia (serum NA < 130 meq/L) was recorded in 12 (16.7%) of the study patients, suggesting that these patients may have had a paraneoplastic syndrome of syndrome of inappropriate antidiuretic hormone (SIADH) or increased ectopic ACTH secretion. Hyponatremia-induced SIADH and ectopic ACTH secretion are paraneoplastic syndromes that commonly occur in SCLC patients, the incidence rate of which is 10–15%. Twelve (16.7%) patients in this study were recorded as having hyponatremia at diagnosis, and this result indicates that the incidence of SIADH or ectopic ACTH secretion was consistent with previous studies [1,2]. The AEs of SIADH and ectopic ACTH secretion are induced by the disease SCLC itself, unrelated to the treatments.

Point 4: Table 2: the patient number of ECOG PS is different from the one on Table 1. Please explain. Also why chose age of 60 as cut-off?

Response 4: We had corrected the error and re-defined the age of 65 as cut-off for analysis in revised manuscript as suggested.

      There was no significant difference in median OS between patients order than 65-year-old and patients’ age less than or equal to 65 (P =0.763).

Point 5: Table3: it is better to expand the adverse events, such as weakness/fatigability, hair loss, mouth sores, etc. which liver transaminase was measured in the study?

Response 5: We expanded the AEs and did more discussion about this point to address this concern as suggested.

Among non-hematological toxicities, the most frequent AE was nausea and vomiting (66.7%), followed by hair loss (54.2%), fatigue (36.1%), diarrhea (18.1%), abnormal renal function (18.1%), increased ALT (16.7%), skin rashes (16.7%), increased AST (15.3%), stomatitis (13.9%), and constipation (2.8%). Only one patient experienced grade 3 diarrhea, which also led to grade 3 elevated serum creatinine. This patient had been hospitalized to receive intravenous fluid supplementation, which corrected the complication of acute kidney injury. This patient did not experience severe diarrhea and kidney injury in the course of subsequent treatment.  

Most grade 3 AEs observed in this study were hematological toxicities. The most com-mon grade 3 hematological AE in this study was neutropenia (22.2%), which led to the treatments being postponed. Twenty-two (30.5%) patients in this study experienced fe-brile neutropenia, and all of the patients received granulocyte colony-stimulating factor (G-CSF) and antibiotics due to concerns about infection. Skin rashes were considered to be an immune-mediated adverse event (imAE); no other severe imAEs—such as pneumonitis, cardiomyopathy, or endocrinopathies—were recorded in this study. Overall, the safety of first-line immunotherapy combined with chemotherapy was manageable, and no treatment-related deaths were recorded in this study.

      AEs were reviewed and analyzed from electronic medical records, and the records of AEs in this study could not be as detailed as those in clinical trials. For example, although stomatitis was noted in this study, the presentations could not be determined as oral pain, sore throat, or mucosal ulceration in oral cavity. Endocrinopathies such as hypothyroidism and adrenal insufficiency sometimes occur in cancer patients who receive anti-PD-L1 ICIs [3], but we did no find these AEs after retrospectively reviewing the electronic medical records.

Point 6: The corresponding author needs to use the institute's email for communication.

Response 6: The email of corresponding author had been changed to his institute's email as suggested.

Reference:

  1. Petereit C, Zaba O, Teber I, Lüders H, Grohé C. A rapid and efficient way to manage hyponatremia in patients with SIADH and small cell lung cancer: treatment with tolvaptan. BMC Pulm Med. 2013 Aug 29;13:55.

  1. Berardi, R.; Mastroianni, C.; Lo Russo, G.; Buosi, R.; Santini, D.; Montanino, A.; Carnaghi, C.; Tiseo, M.; Chiari, R.; Camerini, A.; et al. Syndrome of inappropriate anti-diuretic hormone secretion in cancer patients: results of the first multicenter Italian study. Ther Adv Med Oncol. 2019 Sep 27;11:1758835919877725.

  1. Sosa, A.; Lopez Cadena, E.; Simon Olive, C.; Karachaliou, N.; Rosell, R. Clinical assessment of immune-related adverse events. Ther Adv Med Oncol. 2018 Mar 30;10:1758835918764628.

Reviewer 3 Report

Comments and Suggestions for Authors

Dear Editor and Authors,

Thank you for asking me to review this work titled “A Real-World Analysis of Anti-Programmed Death-Ligand 1 (PD-L1) Immune Checkpoint Inhibitor Treatments Combined with Chemotherapy in Untreated Extensive Stage Small Cell Lung Cancer”.

In this work the authors present a retrospective analysis of a small number of patients diagnosed with advanced stage small cell lung cancer which underwent combination treatment with platinum based chemotherapy and immune checkpoint inhibitor treatment.

Although this study is well written and well presented it has some major methodological flaws which make me very reluctant to recommend its publication. These flaws are so fundamental and core that a revision (unless a redo of the study is performed) would not amend them!!

Specifically:

1.       The study is underpowered!! Only 72 patients were included in the analysis and if one is to consider the other major studies cited (for example the CASPIAN study had almost 550 patients enrolled – with a control group which the authors don’t have!!) the applicability of the findings are questionable!!

2.       I don’t understand the value of utilizing neutrophil-to-lymphocyte ratio (NLR) and serum lactate dehydrogenase (LDH) levels as study end-points? They do not tie together how they feel these values are related to OS and in truth the literature is still debatable as to whether this is true!! See: Bernhardt D, Aufderstrasse S, König L, Adeberg S, Bozorgmehr F, Christopoulos P, El Shafie RA, Hörner-Rieber J, Kappes J, Thomas M, Herth F, Steins M, Debus J, Rieken S. Impact of inflammatory markers on survival in patients with limited disease small-cell lung cancer undergoing chemoradiotherapy. Cancer Manag Res. 2018;10:6563-6569

https://doi.org/10.2147/CMAR.S180990 and

Zhu, J., Lian, L., Qin, H., Wang, W., Ren, R., Xu, M., Chen, K., Duan, W., Gong, F., Tao, M., Zhi, Q., Wu, M., Li, W."Prognostic evaluation of patients with resectable lung cancer using systemic inflammatory response parameters". Oncology Letters 17.2 (2019): 2244-2256.

3.       Most (96%) of patients were male which creates a significant gender bias!!

4.       How were the data recorded and collected? Was there a centralized database with stored variables or was a patient chart review performed? The latter is more plausible but so it has a significant impact on study validity!! Where there missing data?

5.       Although each patient was in an advanced/end stage the organs and degree of metastatic disease varied which affects significantly survival!

6.       Duration and number of therapies varied!!

7.       Some patients also received additional treatment such as radial irradiation which affects PFS and OS!!

8.        The incidence of adverse events seems low considering numbers in the literature!! Has there been a “cherry picking” of outcomes?

9.       How can the authors claim that “The results of our study suggest that additional thoracic radiation therapy may have limited benefit on treatment efficacy or clinical outcomes and should be reserved for patients with paraneoplastic complications such as (SVC) syndrome.” when NO patient received thoracic radiotherapy but brain and bone ones (and those to a rate of 7-15%!!

In conclusion for this reasons I am not able to be supportive of the publication of this work! Kind regards to all.

Comments on the Quality of English Language

Minor language editing is required only.

Author Response

Response to Reviewer 3 Comments

Thank you for asking me to review this work titled “A Real-World Analysis of Anti-Programmed Death-Ligand 1 (PD-L1) Immune Checkpoint Inhibitor Treatments Combined with Chemotherapy in Untreated Extensive Stage Small Cell Lung Cancer”.

In this work the authors present a retrospective analysis of a small number of patients diagnosed with advanced stage small cell lung cancer which underwent combination treatment with platinum based chemotherapy and immune checkpoint inhibitor treatment.

Although this study is well written and well presented it has some major methodological flaws which make me very reluctant to recommend its publication. These flaws are so fundamental and core that a revision (unless a redo of the study is performed) would not amend them!!

Point 1: The study is underpowered!! Only 72 patients were included in the analysis and if one is to consider the other major studies cited (for example the CASPIAN study had almost 550 patients enrolled – with a control group which the authors don’t have!!) the applicability of the findings are questionable!!

Response 1: Regarding the concern about “the study is underpowered”, we added a paragraph in introduction section to explain the aim of this study.

Although atezolizumab or durvalumab in combination with chemotherapy has been suggested as a first-line regimen for the treatment of extensive-stage SCLC, the anti-PD-L1 ICIs are not widely covered by public health insurance policies in most countries, due to concerns about cost-effectiveness [1,2]. The use of atezolizumab or durvalumab is only affordable in a few extensive-stage SCLC patients outside of clinical trials [1,2]. Therefore, the real-world clinical outcomes of extensive-stage SCLC pa-tients receiving combination therapies have rarely been reported, and more clinical experiences of using combination therapies need to be explored. In this study, we aimed to conduct a retrospective multicenter observational study to investigate the efficacy and safety of atezolizumab and durvalumab combined with chemotherapy as first-line treatments for extensive-stage SCLC patients in real-world clinical practice.

Point 2:   I don’t understand the value of utilizing neutrophil-to-lymphocyte ratio (NLR) and serum lactate dehydrogenase (LDH) levels as study end-points? They do not tie together how they feel these values are related to OS and in truth the literature is still debatable as to whether this is true!! See: Bernhardt D, Aufderstrasse S, König L, Adeberg S, Bozorgmehr F, Christopoulos P, El Shafie RA, Hörner-Rieber J, Kappes J, Thomas M, Herth F, Steins M, Debus J, Rieken S. Impact of inflammatory markers on survival in patients with limited disease small-cell lung cancer undergoing chemoradiotherapy. Cancer Manag Res. 2018;10:6563-6569

https://doi.org/10.2147/CMAR.S180990 and

Zhu, J., Lian, L., Qin, H., Wang, W., Ren, R., Xu, M., Chen, K., Duan, W., Gong, F., Tao, M., Zhi, Q., Wu, M., Li, W." Prognostic evaluation of patients with resectable lung cancer using systemic inflammatory response parameters". Oncology Letters 17.2 (2019): 2244-2256. 

Response 2: In response to this concern about that “the value of utilizing neutrophil-to-lymphocyte ratio (NLR) and serum lactate dehydrogenase (LDH) levels as study end-points?”, we discussed this point and cited the 2 publications in revised manuscript.

The NLR has been identified as a predictive factor associated with poor clinical outcomes in patients with NSCLC and SCLC in several previous studies [3-6]. An in-creasing NLR has been proposed to be a response to systemic inflammation, and in previous studies it has been shown to be correlated with the severity of trauma, CVD, and malignancies. These previous studies demonstrated that increasing the NLR negatively affects survival in patients with trauma and cardiovascular events [6, 7]. In our study, the NLR was suggested to be a factor associated with the prognosis of extensive-stage SCLC patients receiving anti-PD-L1 ICIs combined with chemotherapy, but it may not be a biomarker of the treatment response to such therapy. Two previous stud-ies reported that elevated NLR was not identified as an independent factor associated with OS in limited-stage SCLC and resectable NSCLC [8-10]; in contrast to these two studies, all of our study’s patients were at metastatic stages, rather than limited or early stages. A previous study showed that an elevated NLR was associated with increased risk of disease relapse and metastasis in head-and-neck squamous-cell carcinoma [11]

Point 3: Most (96%) of patients were male which creates a significant gender bias!!

Response 3: In response to the concern about “Most (96%) of patients were male which creates a significant gender bias!!”, we had revised our conclusion of this manuscript.

   We added a paragraph in the discussion section in response to this comment.

   Most of the patients in this study were male (95.8%), with only three female patients (4.2%), so our study’s results may be not representative of female extensive-stage SCLC patients. Based on the results of our study, further clinical studies should focus on the efficacy and safety of anti-PD-L1 ICIs combined with chemotherapy in extensive-stage SCLC patients of female sex, ethnic groups other than East Asian, and from different geo-graphic areas.

Point 4: How were the data recorded and collected? Was there a centralized database with stored variables or was a patient chart review performed? The latter is more plausible but so it has a significant impact on study validity!! Where there missing data?

Response 4: Regarding the concern about “How were the data recorded and collected?”, we added a paragraph to describe the collection of study data in method section as suggested. Additional discussion about the limitation of this study was added to address this concern.

The electronic charts of all 79 patients were screened and reviewed by at least 2 of our authors, and 72 patients ultimately met the inclusion criteria of this study. The clinical variables and treatment information from the electronic charts of these 72 study patients were reviewed and recorded for analysis by at least 2 of our authors

      In addition, AEs were reviewed and analyzed from electronic medical records, and the records of AEs in this study could not be as detailed as those in clinical trials. For exam-ple, although stomatitis was noted in this study, the presentations could not be deter-mined as oral pain, sore throat, or mucosal ulceration in oral cavity. Endocrinopathies such as hypothyroidism and adrenal insufficiency sometimes occur in cancer patients who receive anti-PD-L1 ICIs, but we did no find these AEs after retrospectively reviewing the electronic medical records.

Point 5: Although each patient was in an advanced/end stage the organs and degree of metastatic disease varied which affects significantly survival!

Response 5: In response to the comment of “the organs and degree of metastatic disease varied which affects significantly survival!”, we had shown the analysis in table 2 of our revised manuscript.

The factors of distant metastatic sites to brain, bone and liver were not identified to be significantly associated with OS in this study.

Point 6: Duration and number of therapies varied!!

Response 6: Regarding the concern of “duration and number of therapies varied!!”, we had discussed this point in our manuscript.

In the clinical trial IMpower133, the chemotherapy regimen used was carboplatin plus etoposide in all patients. In the other clinical trial, CASPIAN, 78% of the study’s patients received carboplatin plus etoposide as a chemotherapy regimen. In two previous retrospective studies, the treatment regimens were atezolizumab com-bined with carboplatin plus etoposide [12, 13]. In contrast to these previous prospective clinical trials and retrospective studies, most of the patients (68.1%) in our study re-ceived regimens of cisplatin plus etoposide. In Taiwan, the Health Insurance Agency mainly reimburses the chemotherapy regimen cisplatin, rather than carboplatin, for the treatment of small-cell lung cancer [14], which explains why most of the patients in our study received cisplatin as chemotherapy. Fifty-one patients (70.8%) in this study re-ceived durvalumab as immunotherapy, but real-world experience using durvalumab combined with chemotherapy as first-line therapy in extensive SCLC has rarely been reported [15]. Our results showed that patients who received first-line treatment with atezolizumab or durvalumab in combination with chemotherapy experienced no sig-nificant difference in OS. The chemotherapy regimen of cisplatin combined with etoposide is the standard first-line treatment for extensive-stage SCLC [16]. The results of our study indicate that the use of atezolizumab or durvalumab combined with standard chemotherapy regimens as first-line therapy in extensive-stage SCLC is effec-tive and feasible in clinical practice.

Point 7: Some patients also received additional treatment such as radial irradiation which affects PFS and OS!!

Response 7: Regarding the concern of “some patients also received additional treatment such as radial irradiation which affects PFS and OS!!”, we had discussed it deeply in our manuscript.

Due to the retrospective nature of the study, the treatments, including chemotherapy, immuno-therapy, and radiation therapy, were administered based on the judgments of clinical physicians rather than on the protocol of the clinical trial. All 11 patients with baseline brain metastases in our study received palliative radiation therapy during their first-line treatment. In addition, six patients in this study received palliative radiation therapy for bone metastases. The additional local radiation therapy may have positively affected the survival of patients in this study. In the clinical trials IMpower133 and CASPIAN, patients with symptomatic brain metastases were required to be treated be-fore entering the trials. Unlike the prospective clinical trials, some patients in usual clinical practice have symptoms related to brain or bone metastases that need palliative local therapies for symptomatic relief. In the CASPIAN cohort, PCI was administered only in the arm receiving chemotherapy alone, and two patients in this study received PCI after first-line treatment with durvalumab combined with chemo-therapy

Point 8: The incidence of adverse events seems low considering numbers in the literature!! Has there been a “cherry picking” of outcomes?

Response 8: In response to the concern about “The incidence of adverse events seems low considering numbers in the literature!! Has there been a “cherry picking” of outcomes?”, we had expanded the records of AEs in revised manuscript.

Among non-hematological toxicities, the most frequent AE was nausea and vomiting (66.7%), followed by hair loss (54.2%), fatigue (36.1%), diarrhea (18.1%), abnormal renal function (18.1%), increased ALT (16.7%), skin rashes (16.7%), increased AST (15.3%), stomatitis (13.9%), and constipation (2.8%). Only one patient experienced grade 3 diarrhea, which also led to grade 3 elevated serum creatinine. This patient had been hospitalized to receive intravenous fluid supplementation, which corrected the complication of acute kidney injury. This patient did not experience severe diarrhea and kidney injury in the course of subsequent treatment. 

Most grade 3 AEs observed in this study were hematological toxicities. The most common grade 3 hematological AE in this study was neutropenia (22.2%), which led to the treatments being postponed. Twenty-two (30.5%) patients in this study experienced febrile neutropenia, and all of the patients received granulocyte colony-stimulating factor (G-CSF) and antibiotics due to concerns about infection. Skin rashes were considered to be an immune-mediated adverse event (imAE); no other severe imAEs—such as pneumonitis, cardiomyopathy, or endocrinopathies—were recorded in this study. Overall, the safety of first-line immunotherapy combined with chemotherapy was manageable, and no treatment-related deaths were recorded in this study.

Point 9: How can the authors claim that “The results of our study suggest that additional thoracic radiation therapy may have limited benefit on treatment efficacy or clinical outcomes and should be reserved for patients with paraneoplastic complications such as (SVC) syndrome.” when NO patient received thoracic radiotherapy but brain and bone ones (and those to a rate of 7-15%!!

Response 9: In response to this comment, we did further discussion in revised manuscript.

No patient in our study received thoracic radiation therapy during first-line treatment, and the efficacy of first-line immunotherapy combined with chemotherapy in our study was not inferior to that determined in previous studies. The OS outcomes of our study patients were also consistent with those of previous studies [3-7]. The analysis of several previous studies investigating the use of anti-PD-L1 ICIs in extensive-stage SCLC showed that administering thoracic radiation was not positively associated with OS [3-7]. Taking these results together, additional thoracic radiation therapy may have a limited beneficial effect on treatment efficacy or clinical outcomes and should be reserved for patients with paraneoplastic complications such as SVC syndrome.

Reference:

  1. Ionova, Y.; Vuong, W.; Sandoval, O.; Fong, J.; Vu, V.; Zhong, L.; Wilson, L. Cost-Effectiveness Analysis of Atezolizumab Versus Durvalumab as First-Line Treatment of Extensive-Stage Small-Cell Lung Cancer in the USA. Clin Drug Investig. 2022 Jun;42(6):491-500.
  2. Liu, Q.; Luo, X.; Yi, L.; Zeng, X.; Tan, C. First-Line Chemo-Immunotherapy for Extensive-Stage Small-Cell Lung Cancer: A United States-Based Cost-Effectiveness Analysis. Front Oncol. 2021 Jun 29;11:699781.

  1. Qiu, G.; Wang, F.; Xie, X.; Liu, T.; Zeng, C.; Chen, Z.; Zhou, M.; Deng, H.; Yang, Y.; Lin, X.; et al. A retrospective real-world experience of immunotherapy in patients with extensive stage small-cell lung cancer. Cancer Med. 2023 Jul;12(14):14881-14891.
  2. Sato, Y.; Saito, G.; Fujimoto, D. Histologic transformation in lung cancer: when one door shuts, another opens. Ther Adv Med Oncol. 2022 Oct 14;14:17588359221130503.
  3. Qi, W.X.; Xiang, Y.; Zhao, S.; Chen, J. Assessment of systematic inflammatory and nutritional indexes in extensive-stage small-cell lung cancer treated with first-line chemotherapy and atezolizumab. Cancer Immunol Immunother. 2021 Nov;70(11):3199-3206.
  4. Stratmann, J.A.; Timalsina, R.; Atmaca, A.; Rosery, V.; Frost, N.; Alt, J.; Waller, C.F.; Reinmuth, N.; Rohde, G.; Saalfeld, F.C.; et al. Clinical predictors of survival in patients with relapsed/refractory small-cell lung cancer treated with checkpoint inhibitors: a German multicentric real-world analysis. Ther Adv Med Oncol. 2022 Jun 4;14:17588359221097191.
  5. Zhang, Q.; Gong, X.; Sun, L.; Miao, L.; Zhou, Y. The Predictive Value of Pretreatment Lactate Dehydrogenase and Derived Neutrophil-to-Lymphocyte Ratio in Advanced Non-Small Cell Lung Cancer Patients Treated With PD-1/PD-L1 Inhibitors: A Meta-Analysis. Front Oncol. 2022 Jul 18;12:791496.
  6. Chen, Y.H.; Chou, C.H.; Su, H.H.; Tsai, Y.T.; Chiang, M.H.; Kuo, Y.J.; Chen, Y.P. Correlation between neutrophil-to-lymphocyte ratio and postoperative mortality in elderly patients with hip fracture: a meta-analysis. J Orthop Surg Res. 2021 Nov 18;16(1):681.
  7. Bernhardt, D.; Aufderstrasse, S.; König, L.; Adeberg, S.; Bozorgmehr, F.; Christopoulos, P.; Shafie, R.A.E.; Hörner-Rieber, J.; Kappes, J.; Thomas, M.; Herth, F.; et al. Impact of inflammatory markers on survival in patients with limited disease small-cell lung cancer undergoing chemoradiotherapy. Cancer Manag Res. 2018 Nov 30;10:6563-6569.
  8. Zhu, J.; Lian, L.; Qin, H.; Wang, W.J.; Ren, R.; Xu, M.D.; Chen K, Duan W, Gong FR, Tao M, Zhi Q, Wu MY, Li W. Prognostic evaluation of patients with resectable lung cancer using systemic inflammatory response parameters. Oncol Lett. 2019 Feb;17(2):2244-2256.
  9. Mariani, P.; Russo, D.; Maisto, M.; Troiano, G.; Caponio, V.C.A.; Annunziata, M.; Laino, L. Pre-treatment neutrophil-to-lymphocyte ratio is an independent prognostic factor in head and neck squamous cell carcinoma: Meta-analysis and trial sequential analysis. J Oral Pathol Med. 2022 Jan;51(1):39-51.
  10. Lee, S.; Shim, H.S.; Ahn, B.C.; Lim, S.M.; Kim, H.R.; Cho, B.C.; Hong, M.H. Efficacy and safety of atezolizumab, in combination with etoposide and carboplatin regimen, in the first-line treatment of extensive-stage small-cell lung cancer: a single-center experience. Cancer Immunol Immunother. 2022 May;71(5):1093-1101.
  11. Lim, J.U.; Kang, H.S.; Shin, A.Y.; Yeo, C.D.; Kim, S.K.; Kim, J.W.; Kim, S.J.; Lee, S.H. Investigation of poor predictive factors in extensive stage small cell lung cancer under etoposide-platinum-atezolizumab treatment. Thorac Cancer. 2022 Dec;13(23):3384-3392.
  12. Chiang, C.L.; Yang, H.C.; Liao, Y.T.; Luo, Y.H.; Wu, Y.H.; Wu, H.M.; Chen, Y.M. Treatment and survival of patients with small cell lung cancer and brain metastasis. J Neurooncol. 2023 Nov;165(2):343-351.
  13. Qiu, G.; Wang, F.; Xie, X.; Liu, T.; Zeng, C.; Chen, Z.; Zhou, M.; Deng, H.; Yang, Y.; Lin, X.; et al. A retrospective real-world experience of immunotherapy in patients with extensive stage small-cell lung cancer. Cancer Med. 2023 Jul;12(14):14881-14891.
  14. Sato, Y.; Saito, G.; Fujimoto, D. Histologic transformation in lung cancer: when one door shuts, another opens. Ther Adv Med Oncol. 2022 Oct 14;14:17588359221130503.

Round 2

Reviewer 2 Report

Comments and Suggestions for Authors

No more comments.

Reviewer 3 Report

Comments and Suggestions for Authors

Dear Editor and Authors,

Although from the start I was quite unipressed by this study (and I continue to be so) it does not escape me the fact that the authors have made significant improvements in their manuscript and have attempted to a) address all my comments and b) tone done their verbose and exuberant claims.

As such, I am now inclined to agree to publishing this work. It still has significant limittations which due to its nature I don't think the authors can fix. However, they have attempted to inform the readers of those limits and so it is now up to them (the readers) to evaluate the value and accept the findings or reject them! I as a reviewer as mentioned still remain skeptical as to the value of this work so if other reviewers also feel there is no merit in publishing it then I would agree with them easily, however if other reviewers feel this work does have something to offer I do not want to impede it!!

Kind regards to all.

Comments on the Quality of English Language

Minor editing required.